# Mitogenomic Phylogeny of Tonnoidea Suter, 1913 (1825) (Gastropoda: Caenogastropoda)

**DOI:** 10.3390/ani13213342

**Published:** 2023-10-27

**Authors:** Jiawen Zheng, Fengping Li, Mingfu Fan, Zhifeng Gu, Chunsheng Liu, Aimin Wang, Yi Yang

**Affiliations:** 1School of Marine Biology and Fisheries, Hainan University, Haikou 570228, China; jwzheng0806@163.com (J.Z.); 21220951340019@hainanu.edu.cn (F.L.); mingfufan05@163.com (M.F.); hnugu@163.com (Z.G.); lcs5113@163.com (C.L.); aimwang@163.com (A.W.); 2Sanya Nanfan Research Institute, Hainan University, Sanya 572025, China

**Keywords:** Tonnoidean, mitochondrial genome, phylogeny, species diversity

## Abstract

**Simple Summary:**

We sequenced the complete mitochondrial genomes of nine Tonnoidean species, and analyzed the genomic features including genome size, gene order, nucleotide composition and Ka/Ks ratio. The reconstructed phylogeny based on mitogenomic data supported the Tonnoidea classifications at family levels, but their internal relationships remained unclear. At the species level, the present study indicates that species diversity within Bursidae might be underestimated.

**Abstract:**

The Tonnoidea Suter, 1913 (1825) is a moderately diverse group of large predatory gastropods, the systematics of which remain unclear. In the present study, the complete mitochondrial genomes of nine Tonnoidean species were sequenced. All newly sequenced mitogenomes contain 13 protein-coding genes (PCGs), 22 transfer RNA genes and two ribosomal RNA genes, showing similar patterns in genome size, gene order and nucleotide composition. The ratio of nonsynonymous to synonymous of PCGs indicated that NADH complex genes of Tonnoideans were experiencing a more relaxed purifying selection compared with the COX genes. The reconstructed phylogeny based on the combined amino acid sequences of 13 protein-coding genes and the nucleotide sequences of two rRNA genes supported that Ficidae Meek, 1864 (1840) is a sister to Tonnoidea. The monophylies of all Tonnoidean families were recovered and the internal phylogenetic relationships were consistent with the current classification. The phylogeny also revealed that *Tutufa rebuta* (Linnaeus, 1758) is composed of at least two different species, indicating that the species diversity within Bursidae Thiele, 1925 might be underestimated. The present study contributes to the understanding of the Tonnoidean systematics, and it could provide important information for the revision of Tonnoidean systematics in the future.

## 1. Introduction

Tonnoidea Suter, 1913 (1825) is a moderately diverse group of marine predatory gastropods, including nine families (Bursidae Thiele, 1925, Cassidae Latreille, 1825, Charoniidae Powell, 1933, Cymatiidae Iredale, 1913, Laubierinidae Warén & Bouchet, 1990, Personidae Gray, 1854, Ranellidae Gray, 1854, Thalassocyonidae F. Riedel, 1995 and Tonnidae Suter, 1913 (1825)) and 81 genera [1]. Tonnoideans are widely distributed, from intertidal to bathyal areas of all oceans, with most species discovered in tropical and subtropical warm waters [2]. They are known for their ability to secrete sulfuric acid that can be used by the Tonnoideans to prey on various marine invertebrates [3]. Echinoderms appear to constitute the major diet of Tonnoideans [4], although they also feed on other mollusks, polychaetes and sipunculans less frequently [5]. The giant triton snail *Charonia tritonis* (Linnaeus, 1758) is known to prey on crown-of-thorns starfish *Acanthaster planci* (Linnaeus, 1758), and the over-harvesting of *Charonia tritonis* in the Indo-Pacific Ocean could lead to outbreaks of crown-of-thorns starfish that could damage the local coral reefs [6]. Tonnoideans are also known for their long planktonic larval stages [7], with the longest larva period of *Fusitriton oregonensis* (Redfield, 1846) (Cymatiide) recorded as 4.5 years in an aquarium [8]. In Southeast Asia, Tonnoideans are consumed as food, and shells of *Bufonaria rana* (Linnaeus, 1758) have been used in traditional Chinese medicine to cure ulcers and furuncle carbuncles [9]. To better protect and utilize the resources of Tonnoideans, a thorough systematic framework is needed.

The first comprehensive classification of Tonnoidea was established by Thiele [10], and the revisions were made at family and subfamily levels based on Thiele’s framework. Some of the main alterations included the establishment of Laubierinidae and Pisanianurinae and the exclusion of family Ficidae Meek, 1864 (1840) [11]. The latter group was elevated to superfamily Ficoidea based on its special anatomical characters and unique protoconch traits [12]. However, the morphological classification might be prone to the influence of homoplasy and sometimes contradicts molecular phylogenies [13]. Using the concatenated dataset of mitochondrial and nuclear markers, Strong et al. [14] reconstructed a molecular phylogeny of Tonnoidea that were composed of 80 operative taxonomic units within 38 genera. The results indicated that the monophyly of several previously identified groups were not supported. The phylogeny by Strong et al. [14] provided the most thorough framework based on which the family-level classification was revised. Despite the significant results of Strong et al., it is also implied that the short gene fragments were insufficient to resolve deeper phylogenetic relationships of the Tonnoidea derived from the relatively low support values on several internal nodes. Lemarcis et al. [15] provided a mitogenomic scale phylogeny of the neogastropods, including 11 Tonnoideans belonging to five different families. Although their part of the tree containing Tonnoideans had full supports, four of the five families were represented by a single species which meant their monophylies were not able to be determined.

The complete mitochondrial genomes have been widely used in mollusk phylogenetic analyses [16,17,18]. Although some previous studies have reported the flaws of the mitochondrial genome, such as the influence of long-branch attractions in phylogenetic trees [15], it could still provide well-resolved phylogenies below the superfamily level [18]. In the present study, we sequenced the complete mitochondrial genomes of nine Tonnoidea species belonging to four families (Table 1). Together with those mitogenomes published before, the present study aims to provide a robust phylogenetic framework of Tonnoidea and to assess the systematic validity within this superfamily. 

## 2. Materials and Methods

### 2.1. Sample Collection

All the specimens of Tonnoidea were collected along Hainan Island, China (Table 1). After morphological identification, samples were deposited in 95% alcohol in the Laboratory of Economic Shellfish Genetic Breeding and Culture Technology (LESGBCT), Hainan University.

### 2.2. DNA Extraction, Sequencing and Mitogenome Assembly

Total genomic DNA was extracted from foot tissue (about 30 mg) using a TIANamp Marine Animals DNA Kit (Tiangen, Beijing, China) according to the manufacturer’s instructions. The genomic DNA was visualized on more than one 1% agarose gel for quality inspection.

The genomic DNA of all species was sent to Novogene Corporation (Beijing, China) for library construction and next-generation sequencing. The DNA library was generated using NEB Next^®^ Ultra™ DNA Library Prep Kit for Illumina (NEB, Ipswich, MA, USA) according to the manufacturer’s instructions. One library with an insert size of approximately 300 bp was prepared for each species and then sequenced on the Illumina NovaSeq 6000 platform with 150 bp paired-end reads. Finally, about 8 Gb of raw data were generated for each library. After removing the adapters and low-quality reads using Trimmomatic v.0.39 [28], the generated clean reads were imported in Geneious Prime 2021.0.1 [29] for assembly following Irwin et al. [30].

### 2.3. Mitogenome Annotation and Sequence Analysis

Mitochondrial gene annotations were conducted in Geneious Prime. The 13 protein-coding genes (PCGs) were determined using ORF Finder (http://www.ncbi.nlm.nih.gov/orffinder, accessed on 1 August 2023) with the invertebrate mitochondrial genetic code. The secondary structures of transfer RNA (tRNA) genes were predicted by MITOS Webserver [31] and ARWEN [32]. The ribosomal RNA (rRNA) genes were identified by BLAST comparison, and their boundaries were modified according to previously published Tonnoidea mitogenomes. The mitochondrial genome map was generated using CGView [33].

The nucleotide composition and codon usage of PCGs were calculated using MEGA X [34]. The base skew values for a given strand were computed as AT skew = (A − T)/(A + T) and GC skew = (G − C)/(G + C), where A, T, G and C are the occurrences of the four nucleotides. 

The average ratio of nonsynonymous (Ka) to synonymous (Ks) for each gene was calculated using BUSTED analysis [35], implemented on the Datamonkey server (http://www.datamonkey.org/busted, accessed on 1 August 2023) with the genetic code selected as the Invertebrate mitochondrial DNA code and other parameters as default. 

### 2.4. Phylogenetic Analysis 

A total of 21 mitogenomes were selected for phylogenetic analysis, including the nine newly sequenced ones along with seven previously published ones for Tonnoidea (Table 1). To determine the phylogenetic position of Ficidae, the mitogenome of *Ficus variegata* Röding, 1798, and those of two cowries were included. The neogastropod species *Conus consors* G. B. Sowerby I, 1833 and *Conus quercinus* [Lightfoot], 1786 were used as two outgroups.

Compared with the nucleotide sequences of mitochondrial PCGs or nuclear fragments, the amino acid sequences of mitochondrial PCGs have proven to be more efficient to generate better resolved phylogenies between families within Caenogastropoda [18]. Therefore, the amino acid sequences of 13 PCGs and the nucleotide sequences of two rRNA genes were concatenated for phylogenetic reconstruction. The deduced amino acid sequences of the 13 PCGs were translated from the aligned codon sequences, according to the invertebrate mitochondrial genetic code. Nucleotide sequences of the rRNA genes were aligned separately using MAFFT v7 [36] with default parameters. Ambiguously aligned positions were removed using Gblocks v.0.91b [37] with default parameters. Finally, the different single alignments were concatenated into a single dataset in Geneious Prime 2021.0.1. Sequences were converted into different formats for further analyses using DAMBE5 [38].

Phylogenetic relationships were reconstructed based on maximum likelihood (ML) and Bayesian inference (BI) analyses. ML analysis was performed by RAxML-HPC2 on XSEDE [39] with the rapid bootstrap algorithm and 1000 replicates. BI analysis was conducted with MrBayes 3.2.6 [40], running four simultaneous Monte Carlo Markov chains (MCMC) for 10,000,000 generations, sampling every 1000 generations, and discarding the first 25% of the generations as burn-in. Two independent BI runs were carried out to increase the chance of adequate mixing of the Markov chains and to increase the chance of detecting failure to converge. The effective sample size (ESS) of all parameters calculated by Tracer v1.6 was more than 200.

The best partition schemes and best-fit substitution models for the dataset were conducted using PartitionFinder 2 [41], under the Bayesian information criterion (BIC). For the amino acid sequences of 13 PCGs, the partitions tested were all genes combined, all genes separated (except *atp6*-*atp8* and *nad4-nad4L*) and genes grouped by enzymatic complexes (*atp*, *cob*, *cox*, and *nad*). The rRNA genes were analyzed with two different schemes (genes grouped or separated). The best-fit substitution models of the two datasets are provided in Appendix A.

## 3. Results and Discussion

### 3.1. Genome Structure and Composition

The gene annotations of the nine mitogenomes are shown in Appendix A. The mitogenome size of nine species ranged from 15,472 (*Tutufa bubo* (Linnaeus, 1758)) to 15,944 bp (*Phalium flammiferum* (Röding, 1798)), and the differences mainly existed in the non-coding regions. Like the typical metazoan mitogenome [42], the mitogenomes in the present study encode 13 PCGs, 22 tRNA and two rRNA genes, with eight tRNA genes encode in the minor strand while the others encode in the major strand (Figure 1). The gene orders of the newly sequenced mitogenomes are identical to those of Tonnoidean that have been published before [20,25,27]. The nucleotide composition of the nine species is shown in Table 2. The AT content values are from 67.3% to 70.9%, showing a significant AT bias. The higher AT content is a common feature of metazoan mitogenomes [43]. In the nine mitogenomes, AT skew values ranged from −0.111 to −0.093 and GC skews values ranged from 0.015 to 0.075, indicating that the frequencies of A and C on the major chain are lower than T and G, respectively. Similar results have also been found in other invertebrate groups [44,45].

### 3.2. PCGs, tRNA and rRNA Genes

The bias toward a higher representation of the nucleotides A and T in the mitogenomes also leads to a similar bias in almost all PCGs of nine Tonnoidean species, according to the negative AT skew and positive GC skew values. The start and stop codons of the Tonnoideans are shown in Table 3. Most PCGs used the conventional initiation codon ATG, but the *nad4* gene employed an alternative start codon GTG in three species (*Monoplex pilearis* (Linnaeus, 1758), *Tut. bubo* and *Tonna sulcosa* (Born, 1778)). The GTG as the start codon has also been reported in the mitogenomes of other gastropods [46]. The stop codons TAA and TAG were observed in all PCGs except for *ND1*, which employed an incomplete stop codon TA in *M. pilearis* and *Lotoria lotoria* (Linnaeus, 1758). The existence of incomplete stop codons TA or T is very common in metazoan PCGs [47], and truncated stop codons could be transformed into the complete stop codon TAA by post-transcriptional modification [48]. In addition, TAG is present more often than TAA. 

Since the synonymous codon usage bias has been affected by the balance of mutation, selection pressure and genetic drift [49], the analysis of relative synonymous codon usage (RSCU) is important to understand the evolution of mitogenomes. The codons and RSCU represented by *M. pilearis* are shown in Table 4 and Figure 2. Among the nine mitogenomes, UUA is the most frequently used compared with the least-selected ones CGG and CGC. A significant usage bias was found among most amino acids (Figure 2). Furthermore, the bias in synonymous codon usage is also a reflection of base bias of the whole mitogenome, as most of the frequently used codons are composed of bases A and T which correspond to a high AT content for all mitogenomes.

### 3.3. Transfer and Ribosomal RNA Genes

All the Tonnoidean mitogenomes contain 22 tRNA genes, including two copies of tRNA-*Leu*, two of tRNA-*Ser* and one copy of the remaining genes. The secondary structures of 22 tRNA genes represented by *M. pilearis* are shown in Figure 3. Their length ranges from 62 bp (tRNA-*Gln* of *P. glaucum*) to 75 bp (tRNA-*Lys* of *M. pilearis*). However, the same tRNA gene among different mitogenomes shares an almost identical length. 

The 12S rRNA of all Tonnoideans is located between tRNA-*Glu* and tRNA-*Val*, with the length ranging from 956 bp of *G. natator* to 980 bp of *P. flammiferum*. The 16S rRNA is located between tRNA-*Val* and tRNA-*Leu*1, with the length varying from 1361 bp of *L. lotoria* to 1410 bp of *P. flammiferum*. Different from the base bias of the whole mitogenome, the two rRNA genes show positive AT skew values in all species (Table 2). 

### 3.4. Ka/Ks

The Ka/Ks ratios for each gene were all under 1.0, which indicated that all PCGs were under purifying selection (Figure 4). However, the Ka/Ks ratio averaged over all sites and all lineages is almost never >1 since positive selection is unlikely to affect all sites over a prolonged time period [50]. The highest average pairwise Ka/Ks ratio was found in the *atp8* gene, followed by a series of NADH genes (*nad6*, *nad2*, *nad4*, *nad5*, *nad3* and *nad1*) with values ranging from 0.0255 to 0.0817, whereas the lowest Ka/Ks ratio was found in *cox1* with Ka/Ks = 0.0045, followed by *cox2* and *cox3*. This result revealed that the NADH complex genes of Tonnoidean are experiencing a more relaxed purifying selection compared with the COX genes which appear more conservative. Similar trends have also been detected in other gastropod taxa [51,52].

### 3.5. Phylogenetic Relationship 

The dataset used for phylogenetic construction is 5674 bp in length. The best partition scheme for the amino acid sequences of PCGs was combining genes by subunits, while for the nucleotide sequences of rRNA genes, the best scheme was combining 12S and 16S genes. Both ML (−lnL = 52,593.77) and BI (−lnL = 52,216.93 for run1; −lnL = 52,217.15 for run2) arrived at identical topologies (Figure 5). The cowries (family Cypraeidae Rafinesque, 1815) were included in the phylogeny to test the relative position of Ficidae compared with the Tonnoidea. In the present phylogeny, Ficidae represented by the specimen of *F. variegata* was recovered as sister to Tonnoidea, consistent with the phylogenies of Strong et al. [14], Simone [53] and Lemarcis et al. [15]. However, the classification of Ficidae within Tonnoidea has been questioned due to the special anatomical characters and distinctive protoconch of Ficidae [11], indicating that the classification of Ficidae needs to be further investigated with broader taxon sampling. Here we considering Ficidae as a distinct superfamily Ficoidea following the strategy of Strong et al. [14]. The current classification places Tonnoidea within the Littorinimorpha [1]. However, it is worth noting that the latter taxon is likely paraphyletic based on morphological and anatomical phylogeny [54]. Several molecular phylogenies [15,25,27] have also indicated a close relationship between Tonnoidea and Neogastropoda. The precise assignment of Tonnoidea warrants further investigation, but this falls outside the scope of the current study. 

Within Tonnoidea, a total of six families including Personidae, Cymatiidae, Charoniidae, Bursidae, Tonnidae and Cassidae were recognized, with the Personidae branching early within the superfamily (Figure 5). This topology was also recovered by Lemarcis et al. [15]. The comprehensive Tonnoidean systematics indicated that only two genera (*Distorsio* Röding, 1798 and *Personopsis* Beu, 1988) could be assigned within the Personidae. Strong et al. [14] revealed that Personidae was a sister to Thalassocyonidae F. Riedel, 1995, which was not included in the present phylogeny. 

Cymatiidae was the second lineage branching off (Figure 5) the Tonnoidea. Before the Cymatiidae was elevated to the family level, it was defined as a subfamily within Ranellidae. Compared with the original Cymatiinae, the current concept of Cymatiidae has expanded with the inclusion of several genera, including *Argobuccinum* Herrmannsen, 1846, *Fusitriton* Cossmann, 1903 and *Gyrineum* Link, 1807, from subfamily Ranellinae, and with the genus *Charonia* excluded and reassigned to a new family, Charoniidae [14]. The present phylogeny also supported the establishment of Charoniidae due to the separated positions between *Charonia* and Cymatiidae (Figure 5). 

Within the remaining species, the Tonnidae, represented by the single genus *Tonna*, was recovered as a sister to Bursidae + (Charoniidae and Cassidae). Bursidae was known as the frog shell. Morphologically, Bursidae has several unique features, such as a well-defined posterior exhalant siphon at the top of the outer lip and strong shell ornamentation [20]. Bursidae were once believed to possess trans-oceanic dispersal capability due to their extended planktonic larval stage. However, it has been shown that this capability was overestimated for some species within the Bursidae family. For example, the *Dulcerana granularis* (Röding, 1798) was originally considered to be a single species with a worldwide distribution, whereas subsequent systematic analysis revealed that *D. granularis* was composed of at least four endemic species, forming the genus *Dulcerana* Oyama, 1964 [7,20]. A similar result was detected in our phylogeny due to the separated positions of two *Tut. rubeta* individuals from the South China Sea (this study) and Papua New Guinea (GenBank Accession No: MW316790) (Figure 5). In this study, identification of the *Tut. rubeta* specimen was established using both morphological and molecular evidence. Furthermore, all COI fragments accessible in GenBank exhibited an identity value of over 99%, thereby confirming the accuracy of our *Tut. rubeta* identification. Excluding the potential impact of an incorrect species identification for *Tut. rubeta* (MW316790), our results suggest that the diversity of species within Bursidae may have been underestimated. Within Bursidae, *Tutufa* was recovered as sister to *Bufonaria* + *Lampasopsis*, consistent with the studies of Strong et al. [14] and Castelin et al. [55] showing that *Tutufa* was separated from all other bursids. Sanders et al. [20] provided an alternative topology for the inner relationships within Bursidae, whereas there was low support for this part of their tree.

The phylogenetic relationship of the clade grouped by the Bursidae, Charoniidae and Cassidae was not well resolved according to the low support values (Figure 5). On the other hand, the topology of the three families contradicted that of Strong et al. [14], despite the fact that neither was highly supported. However, the present phylogeny of the three families was fully supported by Lemarcis et al. [15]. In Charoniidae, only one genus was included. Cassidae was formed by *Galeodea* and *Phalium + Semicassis*, with the former group belonging to subfamily Cassinae and the latter two to Phaliinae, consistent with the systematics of Strong et al. [14]. However, the phylogeny within this clade needs further study with a broader taxon sampling.

## 4. Conclusions

The present study aims to examine the validity of the current systematics of the Tonnoidea and to provide a robust phylogenetic framework. The complete mitochondrial genomes of nine Tonnoideans sequenced in this study showed similar patterns in genome size, gene order and nucleotide composition. The relatively higher values of the Ka/Ks ratio in NADH complex genes and lower values in COX genes have also been observed in previous studies. The reconstructed mitogenomic phylogeny supported the monophylies of all Tonnoidean families. However, the internal relationships between Cassidae, Charoniidae and Bursidae were not well resolved. The study also indicated that the species diversity within Bursidae might be underestimated and thus calls for further studies with more species included to improve the systematics of Tonnoidea. In parallel to the accumulation of new mitogenomic data, new nuclear markers derived from transcriptomes or reduced genomes also need to be gathered to determine the unresolved internal phylogenetic relationships.

## Figures and Tables

**Figure 1 animals-13-03342-f001:**
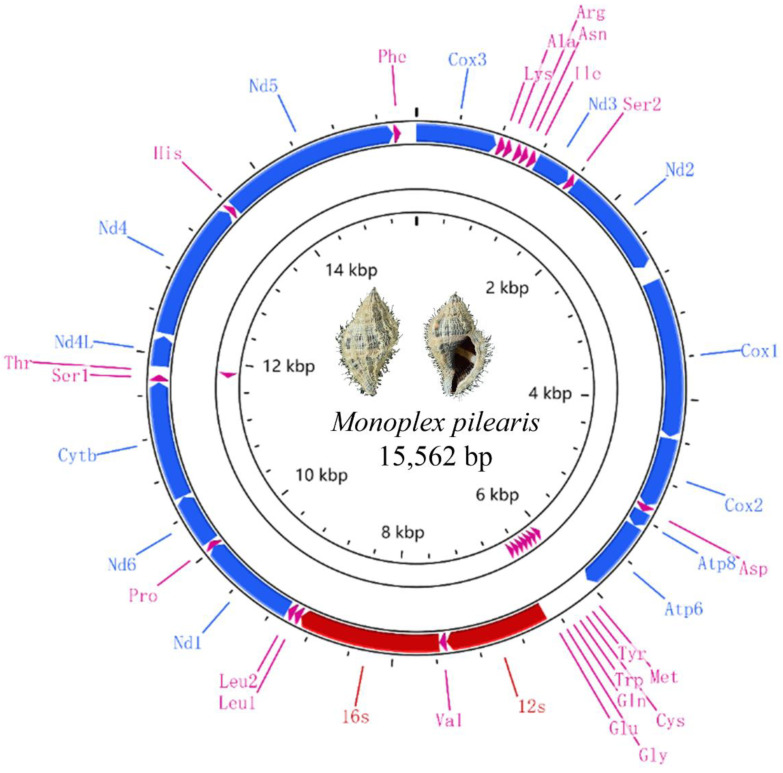
Gene map of the mitogenomes of *Monoplex pilearis*. The blue arrows represent coding genes, the red lines represent rRNA genes and pink arrows represent tRNA genes.

**Figure 2 animals-13-03342-f002:**
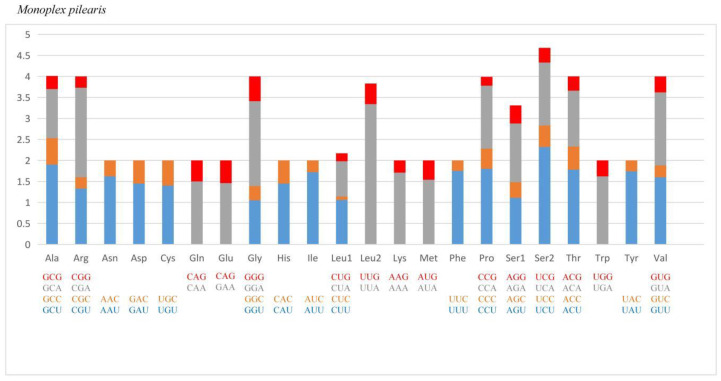
Relative synonymous codon usage (RSCU) of mitochondrial genomes for *Monoplex pilearis*.

**Figure 3 animals-13-03342-f003:**
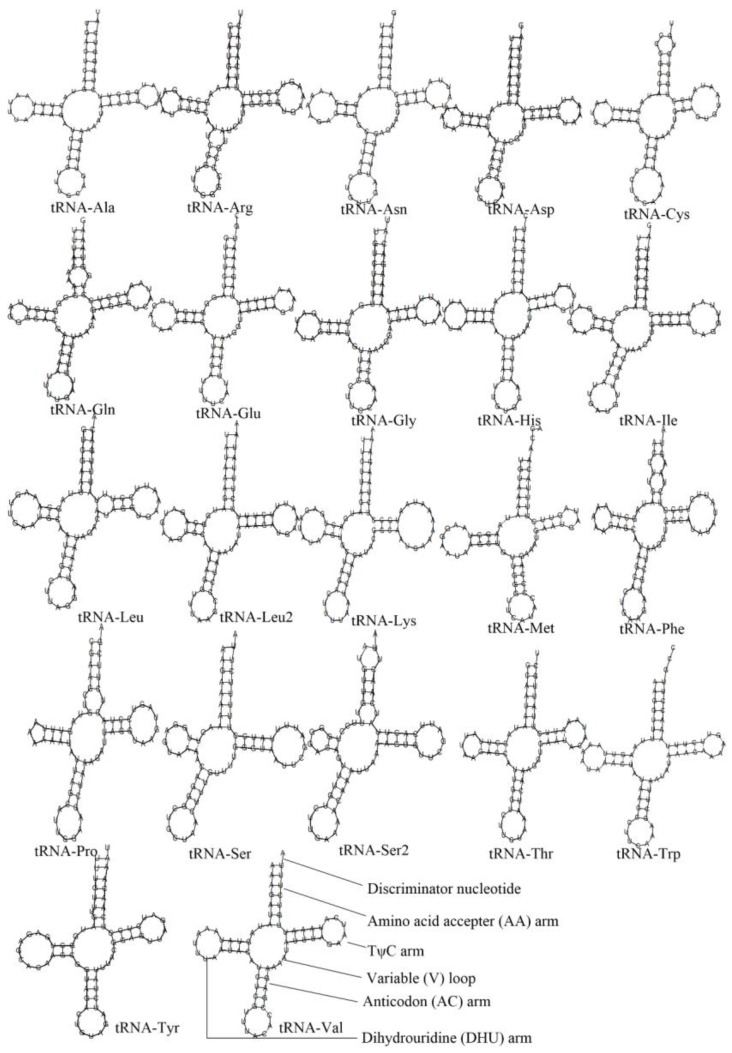
Inferred secondary structures of 22 tRNAs represented by *Monoplex pilearis*. The tRNAs are labeled with their corresponding amino acids. Structural elements in tRNA arms and loops are illustrated as for tRNA-Val.

**Figure 4 animals-13-03342-f004:**
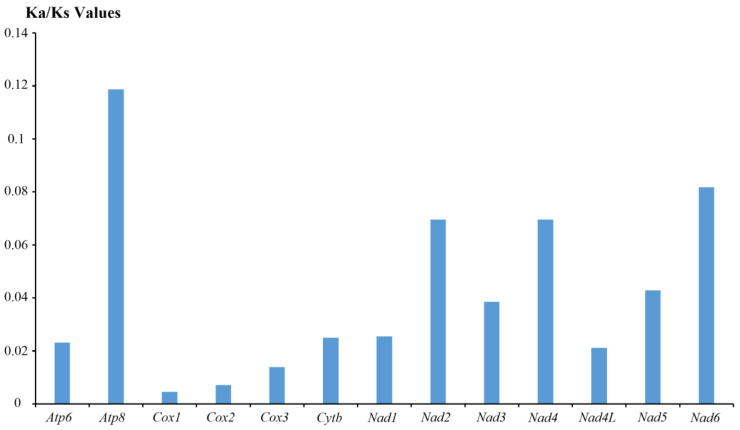
The Ka/Ks ratios of the 13 different mitochondrial genes in Tonnoidean species.

**Figure 5 animals-13-03342-f005:**
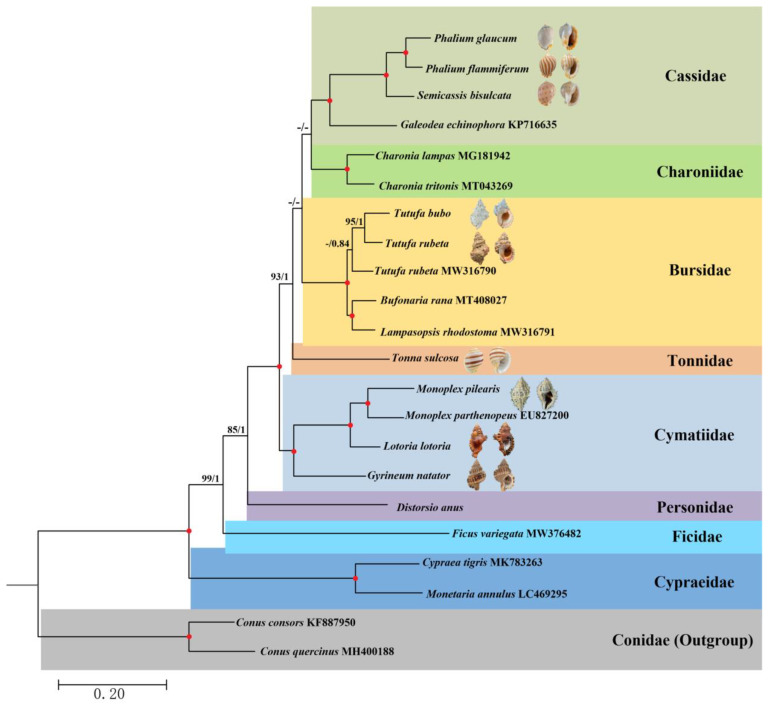
Phylogenetic relationships of Tonnoidea based on the amino acid sequences of 13 mitochondrial protein-coding genes (PCGs) and nucleotide sequences of two ribosomal RNA (rRNA) genes, with *Conus consors* and *C. quercinus* as outgroups. The first number on each node is the bootstrap proportion (BP) of the maximum likelihood (ML) analysis. The second number is the Bayesian posterior probability (PP). Nodes with maximum statistical support (BP = 100; PP = 1) are marked with red solid circles. Only the BP values > 80 and PP values > 0.80 were shown in the tree.

**Table 1 animals-13-03342-t001:** List of the mt genomes analyzed in the present study.

**New mt Genomes**					
**Species**	**Length (bp)**	**Location**	**Institutional Registration Number**	**Date of Sampling**	**Accession Number**
*Monoplex pilearis*	15,562	Sanya, China	LESGBCT001	July 2021	OP689739
*Tutufa bubo*	15,472	Xisha, China	LESGBCT090	September 2021	OP689737
*Gyrineum natator*	15,585	Haikou, China	LESGBCT105	October 2021	OP689738
*Tonna sulcosa*	15,688	Sanya, China	LESGBCT106	November 2021	OP696777
*Lotoria lotoria*	15,821	Sanya, China	LESGBCT109	November 2021	OP696778
*Phalium glaucum*	15,943	Lingshui, China	LESGBCT117	December, 2021	OP696779
*Semicassis bisulcata*	15,833	Sanya, China	LESGBCT134	January 2022	OP696780
*Phalium flammiferum*	15,944	Sanya, China	LESGBCT136	January 2022	OP696781
*Tutufa rubeta*	15,980	Sanya, China	LESGBCT138	January 2022	OP714089
**Genbank mt Genome**					
**Species**	**Length (bp)**	**Accession No.**		**Reference**	
*Bufonaria rana*	15,510	MT408027		Zhong et al., 2020 [19]	
*Lampasopsis rhodostoma*	15,393	MW316791		Sanders et al., 2021 [20]	
*Charonia lampas*	15,330	MG181942		Cho et al., 2017 [21]	
*Charonia tritonis*	15,346	MT043269		Direct Submission	
*Conus consors*	16,112	KF887950		Brauer et al., 2012 [22]	
*Conus quercinus*	16,439	MH400188		Chen et al., 2018 [23]	
*Cypraea tigris*	16,177	MK783263		Pu et al., 2019 [24]	
*Ficus variegata*	15,736	MW376482		Direct Submission	
*Galeodea echinophora*	15,388	KP716635		Osca et al., 2015 [25]	
*Monetaria annulus*	16,087	LC469295		Fukumori et al., 2019 [26]	
*Monoplex parthenopeus*	15,270	EU827200		Cunha et al., 2009 [27]	
*Tutufa rubeta*	15,397	MW316790		Sanders et al., 2021 [20]	

**Table 2 animals-13-03342-t002:** Total size, AT content, AT skew and GC skew for mitochondrial genomes of *Monoplex pilearis*, *Tutufa bubo*, *Gyrineum natator*, *Tonna sulcosa*, *Phalium glaucum*, *Lotoria lotoria*, *Semicassis bisulcata*, *Phalium flammiferum* and *Tutufa rubeta*.

	*Monoplex pilearis*	*Tutufa bubo*	*Gyrineum natator*	*Tonna sulcosa*	*Phalium glaucum*	*Lotoria lotoria*	*Semicassis bisulcata*	*Phalium flammiferum*	*Tutufa rubeta*
%A+T	69%	67.30%	68.80%	70.90%	70.60%	69.10%	69.90%	70.30%	67.70%
AT skew (mt genome)	−0.110	−0.103	−0.110	−0.111	−0.093	−0.111	−0.107	−0.101	−0.099
AT skew (PCGs)	−0.170	−0.154	−0.163	−0.159	−0.150	−0.167	−0.170	−0.164	−0.163
AT skew (rRNAs)	0.071	0.064	0.052	0.056	0.078	0.048	0.066	0.072	0.076
GC skew (mt genome)	0.052	0.015	0.029	0.075	0.031	0.055	0.056	0.051	0.022
GC skew (PCGs)	0.032	−0.018	0.006	0.056	0.003	0.035	0.039	0.026	0
GC skew (rRNAs)	0.138	0.171	0.146	0.163	0.164	0.154	0.144	0.157	0.167

**Table 3 animals-13-03342-t003:** Length and start/stop codon of protein-coding genes in nine species.

	*Monoplex pilearis*	*Tutufa bubo*	*Gyrineum natator*	*Tonna sulcosa*	*Phalium glaucum*	*Lotoria lotoria*	*Semicassis bisulcata*	*Phalium flammiferum*	*Tutufa rubeta*
Total	15,562	15,472	15,585	15,688	15,943	15,821	15,833	15,944	15,980
rrnS	979	972	956	961	968	960	964	980	969
rrnL	1368	1394	1384	1397	1408	1361	1383	1410	1396
*Atp6*	696(ATG/TAA)	696(ATG/TAG)	696(ATG/TAA)	696(ATG/TAA)	696(ATG/TAA)	699(ATG/TAA)	696(ATG/TAA)	696(ATG/TAA)	696(ATG/TAG)
*Atp8*	159(ATG/TAA)	159(ATG/TAA)	159(ATG/TAA)	159(ATG/TAA)	159(ATG/TAA)	159(ATG/TAA)	159(ATG/TAA)	159(ATG/TAA)	159(ATG/TAA)
*cob*	1140(ATG/TAA)	1140(ATG/TAA)	1140(ATG/TAA)	1140(ATG/TAA)	1140(ATG/TAA)	1140(ATG/TAA)	1140(ATG/TAA)	1140(ATG/TAA)	1140(ATG/TAA)
*Cox1*	1536(ATG/TAA)	1536(ATG/TAG)	1536(ATG/TAA)	1536(ATG/TAA)	1536(ATG/TAA)	1536(ATG/TAA)	1536(ATG/TAA)	1536(ATG/TAA)	1536(ATG/TAA)
*Cox2*	687(ATG/TAA)	687(ATG/TAA)	687(ATG/TAA)	687(ATG/TAA)	687(ATG/TAA)	687(ATG/TAA)	687(ATG/TAA)	687(ATG/TAA)	687(ATG/TAA)
*Cox3*	780(ATG/TAA)	780(ATG/TAG)	780(ATG/TAA)	780(ATG/TAG)	780(ATG/TAA)	780(ATG/TAA)	780(ATG/TAA)	780(ATG/TAA)	780(ATG/TAG)
*Nad1*	941(ATG/TA)	942(ATG/TAA)	942(ATG/TAA)	942(ATG/TAG)	942(ATG/TAA)	941(ATG/TA)	942(ATG/TAA)	942(ATG/TAA)	942(ATG/TAA)
*Nad2*	1059(ATG/TAA)	1059(ATG/TAA)	1059(ATG/TAA)	1059(ATG/TAA)	1059(ATG/TAA)	1059(ATG/TAG)	1059(ATG/TAA)	1059(ATG/TAG)	1059(ATG/TAA)
*Nad3*	354(ATG/TAA)	354(ATG/TAA)	354(ATG/TAA)	354(ATG/TAG)	354(ATG/TAA)	354(ATG/TAG)	354(ATG/TAA)	354(ATG/TAA)	354(ATG/TAA)
*Nad4*	1374(GTG/TAA)	1374(GTG/TAA)	1374(ATG/TAA)	1377(GTG/TAA)	1374(ATG/TAG)	1374(ATG/TAA)	1374(ATG/TAA)	1374(ATG/TAA)	1374(ATG/TAA)
*Nad4L*	297(ATG/TAG)	297(ATG/TAG)	297(ATG/TAG)	297(ATG/TAG)	297(ATG/TAG)	297(ATG/TAG)	297(ATG/TAG)	297(ATG/TAG)	297(ATG/TAG)
*Nad5*	1722(ATG/TAA)	1722(ATG/TAG)	1722(ATG/TAG)	1722(ATG/TAG)	1722(ATG/TAA)	1722(ATG/TAA)	1722(ATG/TAA)	1722(ATG/TAA)	1722(ATG/TAG)
*Nad6*	501(ATG/TAG)	501(ATG/TAA)	495(ATG/TAG)	501(ATG/TAA)	501(ATG/TAA)	501(ATG/TAA)	501(ATG/TAA)	501(ATG/TAA)	501(ATG/TAA)

**Table 4 animals-13-03342-t004:** Codon and relative synonymous codon usage (RSCU) of 13 PCGs in the mt genomes of *Monoplex pilearis*.

Amino Acid	Codon	Count	RSCU	Amino Acid	Codon	Count	RSCU
Phe	UUU(F)	285	1.75	Tyr	UAU(Y)	122	1.74
	UUC(F)	40	0.25		UAC(Y)	18	0.26
Leu	UUA(L)	320	3.34	His	CAU(H)	64	1.45
	UUG(L)	47	0.49		CAC(H)	24	0.55
	CUU(L)	101	1.06	Gln	CAA(Q)	60	1.50
	CUC(L)	8	0.08		CAG(Q)	20	0.50
	CUA(L)	80	0.84	Asn	AAU(N)	90	1.62
	CUG(L)	18	0.19		AAC(N)	21	0.38
Ile	AUU(I)	261	1.72	Lys	AAA(K)	77	1.71
	AUC(I)	42	0.28		AAG(K)	13	0.29
Met	AUA(M)	166	1.54	Asp	GAU(D)	58	1.45
	AUG(M)	49	0.46		GAC(D)	22	0.55
Val	GUU(V)	97	1.60	Glu	GAA(E)	62	1.46
	GUC(V)	17	0.28		GAG(E)	23	0.54
	GUA(V)	105	1.74	Cys	UGU(C)	28	1.40
	GUG(V)	23	0.38		UGC(C)	12	0.60
Ser	AGU(S)	54	1.11	Trp	UGA(W)	90	1.62
	AGC(S)	18	0.37		UGG(W)	21	0.38
	AGA(S)	68	1.40	Arg	CGU(R)	20	1.33
	AGG(S)	21	0.43		CGC(R)	4	0.27
	UCU(S)	113	2.32		CGA(R)	32	2.13
	UCC(S)	25	0.51		CGG(R)	4	0.27
	UCA(S)	73	1.50	Pro	CCU(P)	67	1.80
	UCG(S)	17	0.35		CCC(P)	18	0.48
Thr	ACU(T)	78	1.78		CCA(P)	56	1.50
	ACC(T)	24	0.55		CCG(P)	8	0.21
	ACA(T)	58	1.33	Gly	GGU(G)	64	1.05
	ACG(T)	15	0.34		GGC(G)	21	0.34
Ala	GCU(A)	112	1.90		GGA(G)	123	2.02
	GCC(A)	37	0.63		GGG(G)	36	0.59
	GCA(A)	69	1.17	-	UAA	10	1.67
	GCG(A)	18	0.31		UAG	2	0.33

## Data Availability

All the newly sequenced genomic data in this study are deposited in the GenBank database with accession numbers OP689737-OP689739, OP696777-OP696781 and OP714089.

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
