# Peer review of "Mitogenomic Phylogeny of Tonnoidea Suter, 1913 (1825) (Gastropoda: Caenogastropoda)"

_animals, 2023, doi:10.3390/ani13213342_

Round 1

Reviewer 1 Report

Comments and Suggestions for Authors

The authors present an interesting study on mitochondrial genome structure and phylogenetic implications of tonnoidean species. The bright side of the manuscript is that to provide practical details to study such species in this content, structure of mitochondrial genome and phylogenetic relationship of the species. Only minor concerns are raised. Therefore, I would like to make some suggestions to improve the quality of the paper as below:

Line 10: “Simple Summary:” should be bold.

Line 16: I think, it is better to say “unclear”, “unsolved” instead of “open”.

Lines 26-27: “The present study could provide important information for the revision of tonnoidean systematics in the future.” -> “The present study contributes to the understanding of the tonnoidean systematics, and it could provide important information for the revision of tonnoidean systematics in the future.”

Line 78: “according to instructions” -> “according to manufacturer's instructions”.

Line 152: Monoplex pilearis -> Monoplex pilearis

Lines 196-198: Table 2: species names should be italic.

Lines 263-270: The conclusion should be rephrased as follows; please start with a brief description of the study (the aim of the study with a sentence), explain the main findings of the study briefly (the results that the authors found), explain how your results contribute to tonnoidean systematics with 2-3 sentences, and explain the limitations of the study and describe the future remarks briefly.

Author Response

Dear Reviewer,

We really appreciate for your comments on our manuscript. We have studied the comments carefully and made corrections in this manuscript. The responses are shown below.

Point1: Line 10: “Simple Summary:” should be bold.

Response1: It was revised as bold.

Point2: Line 16: I think, it is better to say “unclear”, “unsolved” instead of “open”.

Response2: We agree with this point, and used “unclear” to replace “open”.

Point3: Lines 26-27: “The present study could provide important information for the revision of tonnoidean systematics in the future.” -> “The present study contributes to the understanding of the tonnoidean systematics, and it could provide important information for the revision of tonnoidean systematics in the future.”

Response3: We have revised this sentence as the Reviewer suggested.

Point4: Line 78: “according to instructions” -> “according to manufacturer's instructions”.

Response4: This sentence was revised following the Reviewer’s adivice.

Point5: Line 152: Monoplex pilearis -> Monoplex pilearis

Response5: The species name has been changed as italic.

Point6: Lines 196-198: Table 2: species names should be italic.

Response6: All the species names were revised as italic in Table2.

Point7: Lines 263-270: The conclusion should be rephrased as follows; please start with a brief description of the study (the aim of the study with a sentence), explain the main findings of the study briefly (the results that the authors found), explain how your results contribute to tonnoidean systematics with 2-3 sentences, and explain the limitations of the study and describe the future remarks briefly.

Response7: This comment is valuable. Following the instruction of the Reviewer, we have rephrased the conclusion part as following.

“The present study aims to examine the validity of the current systematics of the Tonnoidea and provide a robust phylogenetic framework. The complete mitochondrial genomes of nine tonnoideans sequenced in this study showed similar patterns in genome size, gene order and nucleotide composition. The relative higher values of Ka/Ks ratio in NADH complex genes and lower values in COX genes have also been observed in previous studies. The reconstructed mitogenomic phylogeny supported the monophylies of all tonnoidean families. However, the internal relationships between Cassidae, Charoniidae and Bursidae were not well-resolved. The study also indicated that the species diversity within Bursidae might be underestimated, and called for further studies with more species included to improve the systematics of Tonnoidea. In parallel to the accumulation of new mitogenomic data, new nuclear markers derived from transcrip-tomes or reduced genomes need also to be gathered to determine the unresolved internal phylogenetic relationships.”

Reviewer 2 Report

Comments and Suggestions for Authors

Author Response

Dear Reviewer,

We really appreciate for your valuable comments on our manuscript. We have studied the comments carefully and made corrections in this manuscript. The responses are shown below.

Point1: The biggest trouble with this paper is that the authors missed the paper Neogastropod (Mollusca, Gastropoda) phylogeny: A step forward with mitogenomes by Lemarcis et al., 2022. This paper presents a mitogenomic scale phylogeny including tonnoideans. Not knowing it is an impediment for the discussion.

Response1: This comment is quite important. Actually we already noticed this important work by Lemarcis et al. published in the past year. The reason that it was not cited is because we reconstructed our phylogeny before it was published. Following the advice of the Reviewer, we have included this work for the discussion in the revised version.

Point2: Furthermore, this study provided tonnoidean mitogenomes from a family not included in the present paper (Personidae). The authors must re-conduct the phylogenetic analyses with the inclusion of the distorsio anus mitogenome (missing 12s and 16s) available here: (https://www.ncbi.nlm.nih.gov/nuccore/?term=danu2). Doing so, it will very probably change the discussion and conclusion.

Response2: The mitogenomic data of Distorsio anus was included for phylogenetic reconstruction in the latest version. And the figure 5 as well as corresponding descriptions in the discussion and conclusion parts were therefore changed.

Point3: Authors must provide institutional number for the nine vouchers.

Response3: The institutional number for the nine vouchers was added in Table 1.

Point4: If good quality pictures of the vouchers could be provided as supplementary material it would be appreciated.

Response4: We really apologize since we were not able to provide more pictures due to limitation of equipment.

Point5: Authors must be careful of the proper naming of taxonomic units name author and date in full when first mentioned and with abbreviation without author and date subsequently.

Response5: Following this advice, we have revised all the taxonomic units name throughout the manuscript.

Point6: Mitochondrial and nuclear markers could have different phylogenetic signal; elaborate on your choice not to include nuclear markers such as H3, 18s or 28s in your analysis

Response6: We admit that nuclear fragments and mitogenomic data could arrive at different topologies, however, a previous study indicated that the amino acid sequences of mitochondrial PCGs were more efficient to generate better resolved phylogenies between families within the same superfamily of Caenogastropoda (Uribe et al., 2017 https://doi.org/10.1016/j.ympev.2016.10.008). Therefore we followed this work and performed a dataset containing the amino acid sequences of 13 PCGs and the nucleotide sequences of 2 rRNA genes to reconstruct the phylogenetic relationships of different families within Tonnoidea.

Point7: Tonnoidea should be followed by Suter, 1913 (1825) in title

Response7: The author name and date were added in title.

Point8: I would advised against using Littorinimorpha as it is most likely a paraphyletic group (see Strong 2003: https://doi.org/10.1046/j.1096-3642.2003.00058.x), You could use Caenogastropoda instead)

Response8: We agree with the Reviewer, and we used Caenogastropoda to replace Littorinimorpha.

Point9: Summary should be in bold

Response9: It was revised as bold.

Point10: genomes of nine Tonnoidea species → should be nine tonnoidean species or nine species of Tonnoidea Suter, 1913 (1825)

Response10: It was changed to “nine tonnoidean species”.

Point11: Tonnoidea should be followed by Suter, 1913 (1825) when first mentioned in the abstract

Response11: The author name and date of Tonnoidea were added in the abstract.

Point12: L17: nine Tonnoidea species → sould be “nine tonnoidean species”

Response12: Revised to “nine tonnoidean species”

Point13: L23: that Ficidae as sister to Tonnoidea → that Ficidae Meek, 1864 (1840) as sister group to Tonnoidea

Response13: The author name and date of Ficidae were added

Point14: L25: The phylogeny also revealed Tutufa rebuta included at least two different species → The phylogeny also revealed that Tutufa rubeta (Linnaeus, 1758) is at least composed of two different species

Response14: We accepted this advice and made the revision.

Point15: L26: Bursidae should be Bursidae Thiele, 1925 when first mentioned in the abstract

Response15: The author name and date of Bursidae were added

Point16: L31: Tonnoidea should be followed by Suter, 1913 (1825) when first mentioned in the text.

Response16: The author name and date of Tonnoidea were added

Point17: L32: You should indicate all families by name.

Response17: The family names were indicated.

Point18: L35-36: « However, echinoderms appearto constitute the major diet of Tonnoideans », you should add that they also feed on other mollusks, polychaetes and sipunculans even though it’s less common.

Response18: We accepted this advice and rephrased the sentence as “Echinoderms appear to constitute the major diet of tonnoideans [4], although they also feed on other mollusks, polychaetes and sipunculans which are less common [5]”.

Point19: L37: Acanthaster planci Acanthaster planci (Linnaeus, 1758)

L38: Charonia tritonisCharonia tritonis (Linnaeus, 1758)

L40: Fusitriton oregonensis Fusitriton oregonensis Redfield, 1846

Response19: The author names and dates of these species were added.

Point20: L41: (Ranellidae): wrong Fusitriton oregonensis belong to Cymatiidae Iredale, 1913 (1854), see strong et al., 2018

Response20: We agree with this point and made the revision.

Point21: L42: Bufonaria ranaBufonaria rana (Linnaeus, 1758)

Response21: The author name and date were added.

Point22: L46: based on the framework of Thiele → Based on Thiele’s framework

L47: Some of the major changes, for example → Some of the main alterations included…

Response22: We agree and changed the sentences.

Point23: L48: Ficidae → Ficidae Meek, 1864 (1840)

Response23: The author name and date were added.

Point24: L50-51: “However, the morphological classification might be prone to the influence of homoplasy, and sometimes contradicts molecular phylogenies”. A citation should support this statement

Response24: We have cited a previous study by Galindo et al. (2016) https://doi.org/10.1016/j.ympev.2016.03.019 to support this statement.

Point25: L53-54: “that recognized about 80 species and 38 genera.” The phylogeny of strong et al. did not recognized 80 species. It was composed of 80 OUT (operative taxonomic units) attributed to the species level, within 38 genera

Response25: We have noticed this error, and changed “80 species” to “80 operative taxonomic units”.

Point26: L59: at this point you should cite and comment Lemarcis et al. 2022 (https://doi.org/10.1111/zsc.12552) they provided a mitogenomic scale phylogeny of the neogastropods, including 11 tonnoideans including the mitogenome of Distorsio anus (Linnaeus, 1758), a personid not included in your phylogeny. Their part of the tree containing tonnoideans have full support.

Response26: We have cited the work of Lemarcis et al. 2022, and made comments as following.

“Lemarcis et al. [15] provided a mitogenomic scale phylogeny of the neogastropods, in-cluding 11 tonnoideans belonging to five different families. Although their part of the tree containing tonnoideans had full supports, four of the five families were represented by a single species which meant their monophylies were not able to be determined.”

Point27: L60-62: this sentence should be changed in the light of Lemarcis et al.,

Response27: This sentence was rephrased as “The complete mitochondrial genomes have been widely used in mollusk phylogenetic analyses [16-18]. Although some previous studies have reported the flaws of the mito-chondrial genome, such as the influence of long-branch attractions in phylogenetic trees [15], it could still provide well-resolved phylogenies below the superfamily level [18].”

Point28: Table1 :Provide institutional registration numbers for the nine new specimens, it is for repeatability and reproducibility purposes.

Response28: The institutional registration numbers for the nine new specimens were added in Table 1.

Point29: Charonia lampas mitogenome was produced by Cho et al. 2017 (https://doi.org/10.1080/23802359.2017.1398610) not Sanders et al.

Response29: We have revised this error.

Point30: In tables I would recommend the addition of author and dates after species names however it’s not mandatory. Though, Ficus variegata Röding should be Ficus variegata Röding, 1798 or just Ficus variegate

Response30: Since the place is quite limited in the table, we decide not to put the author names and dates after species. However, the author name “Röding” was deleted.

Point31: Zhong et al., 2020, Brauer et al., 2012, Chen et al., 2018, Pu et al., 2019, Fukumori et al., 2019 and Cunha et al., 2009 are missing from the reference list.

Response31: They were added in the reference list.

Point32: The correct form to cite a publication with more than two authors is et al., in italic with a point before the coma.

Response32: We have normalized “et al.” as suggested.

Point33: Please specify the difference between Unpublished and Direct Submission

Response33: “Unpublished” should be replaced by “Direct Submission”.

  1. Material and Methods

Point34: L103: specify the parameters used in Busted (Genetic Code, Synonymous rate variation (recommended), Advanced Include support for multiple nucleotide substitutions).

Response34: The genetic code was selected as the Invertebrate mitochondrial DNA code and other parameters as default. 

Point35: L104: http://www.datamonkey.org/dataupload.php is not a valid url, should be http://www.datamonkey.org/busted

Response35: It was revised.

Point36: L108: Ficus variegataFicus variegata Röding, 1798

L109: Conus consorsConus consors G. B. Sowerby I, 1833; C. quercinus Conus quercinus [Lightfoot], 1786

  1. Results and Discussion

L138: Tutufa buboTutufa bubo (Linnaeus, 1758) and Phalium flammiferumPhalium flammiferum (Röding, 1798)

Response36: The author names and dates were added.

Point37: L143: “tonnoidean published before [31]”. You should cite more than on paper, I suggest you add Cunha et al., 2009, Osca et al., 2015 and/or Sanders et al., 2021. Those papers (including the one already cited in the text) discussed genes order for tonnoideans from three different families.

Response37: We have cited two more papers as suggested.

Point38: L158: Tutufa and Tonna should not have the same abbreviation; M. pilearisMonoplex pilearis (Linnaeus, 1758) as it is the first time it is mentioned in the main text

L161: L. lotoria Lotoria lotoria (Linnaeus, 1758) as it is the first time it is mentioned in the main text

Response38: The author names and dates were added. We used the first three letters to distinguish the two genera.

Point39: Figure 1: Please indicate that blue arrows represent coding genes, red lines represent rRNA genes, and pink arrows represents tRNA. Monoplex pilearis should be in italic

Response39: The caption of figure 1 was revised.

Point40: Figure 2 Monoplex pilearis should be in italic

Table 2. Species names should be in italic in the title. This table occurs after table 3 and 4 in the text

Response40: The species names was revised as italic in the title of Figure 2 and Table 3. The location of Table 2 was changed to the place where it was firstly mentioned.

Point41: Figure 4. This figure has very little use. I do not understand the choice of a line graph instead of a histogram or even a plane table presenting the Ka/Ks values.

Response 41: We revised the Figure 4 from line graph style to histogram style.

Point42: L218: Cypraeidae → Cypraeidae Rafinesque, 1815

Response 42: The author name and date were added

Point43: L220: Ficidae represented by Ficus → Ficidae represented by the specimen of F. variegate

Response 43: Revised as suggested.

Point44: L221: “consistent with the phylogenies of Strong et al. [12] and Simone [42].” It is also the case in Lemarcis et al. 2022

Response 44: We agreed with this point, and cited Lemarcis et al. (2022).

Point45: Figure 5. The support value representation is not consistent, sometime you have -/PP and some other time it’s 100/PP. if you stick with the -/PP representation indicate its meaning in the legend

Response 45: The “-” indicates BP values < 80 and PP values < 0.80. We have added the description in the legend.

Point46: L234: “a total of five families were recognized” name them.

Response 46: The names were listed.

Point47: L234/235: “the Cymatiidae as a basal position (Figure 5)”. Basal as a gradist connotation, a better formulation would be: with the Cymatiidae branching early within the superfamily.

Response 47: Revised as suggested.

Point48: L237/238: “Cymatiidae has expanded with the inclusion of several genera from subfamily Ranellinae” give example(s).

Response 48: Three genera including Argobuccinum Herrmannsen, 1846, Fusitriton Cossmann, 1903 and Gyrineum Link, 1807 were listed in the manuscript.

Point49: L246: “the Bursa granular was” → Bursa granularis (Röding, 1798) (now accepted as Dulcerana granularis)

Response 49: Revised as suggested.

Point50: L248: revealed that Bursa granular was comprised of at least four endemic species [6] → revealed that Bursa granularis was comprised of at least four endemic species forming the genus Dulcerana Oyama, 1964 [6]&[43].

Response 50: Revised as suggested.

Point51: L249: Tutufa rebutaTutufa rubeta

Response 51: Revised as suggested.

Point52: L249/251 “result was detected in our phylogeny due to the separated positions of two Tutufa rebuta individuals from the South China Seas (this study) and Papua New Guinea (GenBank Accession No: MW316790), respectively (Figure 5), indicating that species diversity within Bursidae might be underestimated”. Before saying that biodiversity may be underestimated, you should consider a possible error of identification (from your part or Sanders et al.’s), there is 13 species of Tutufa recognized today and only 4 (5 if you’re right) for which we have molecular data. I understand that concluding may be complicated but it should be discussed anyhow.

Response 52: The identification of Tut. rubeta specimen in this study was conducted based on morphological and molecular evidence. And all the COI fragments available on GenBank showed an identity value of more than 99%, supporting the validity of our Tut. rubeta specimen. From this point, the possible error of species identification could be eliminated from our part. We noticed that the published COI sequences which were almost identical to ours were from the same institution of Sanders et al., which means these samples of Tut. rubeta should be quite available for Sanders et al.. And thus, we assumed that identification error could also be excluded from Sanders et al.’s part.

Point53: L253-254: You should discussed the results of Sanders et al., [43] as they provide an alternative topology for the inner relationships within Bursidae. It could be as simple as to say there was low support for this part of the tree.

Response 53: We have mentioned the alternative topology of Sanders et al. [43], and commented there was low support for this part of the tree as the Reviewer suggested.

Point54: 255-270: in the light of the inclusion of Distorsio anus to the analysis this sections should be rewritten.

Response 54: With the inclusion of Distorsio anus, the topology of Tonnoidea has changed. Therefore we have rewritten this section.

References:

Point55: L356: DOI is missing for Osca et al., 2015 → https://doi.org/10.1016/j.ympev.2015.07.011

Response 55: The DOI was added for Osca et al., 2015.

Point56: L379 : Arquivos de Zoologia 2011, 42, → Arquivos de Zoologia 2011, 42(4),

Response 56: Revised as suggested.